# Implementation of Engagement Detection for Human–Robot Interaction in Complex Environments

**DOI:** 10.3390/s24113311

**Published:** 2024-05-22

**Authors:** Sin-Ru Lu, Jia-Hsun Lo, Yi-Tian Hong, Han-Pang Huang

**Affiliations:** Mechanical Engineering Department, National Taiwan University, Taipei 10617, Taiwan; r07522801@ntu.edu.tw (S.-R.L.); d11522006@ntu.edu.tw (J.-H.L.); r12522813@ntu.edu.tw (Y.-T.H.)

**Keywords:** action recognition, cognitive system, engagement, human–robot interaction, human behaviors

## Abstract

This study develops a comprehensive robotic system, termed the robot cognitive system, for complex environments, integrating three models: the engagement model, the intention model, and the human–robot interaction (HRI) model. The system aims to enhance the naturalness and comfort of HRI by enabling robots to detect human behaviors, intentions, and emotions accurately. A novel dual-arm-hand mobile robot, Mobi, was designed to demonstrate the system’s efficacy. The engagement model utilizes eye gaze, head pose, and action recognition to determine the suitable moment for interaction initiation, addressing potential eye contact anxiety. The intention model employs sentiment analysis and emotion classification to infer the interactor’s intentions. The HRI model, integrated with Google Dialogflow, facilitates appropriate robot responses based on user feedback. The system’s performance was validated in a retail environment scenario, demonstrating its potential to improve the user experience in HRIs.

## 1. Introduction

The human–robot interaction (HRI) attempts to shape the interaction between one or more individuals and one or more robots. The integration of robotics into human-centric environments necessitates a nuanced understanding of HRI to ensure safety, efficiency, and user satisfaction. Recent studies have made significant strides in this domain, addressing cognitive ergonomics in collaborative robotics [1], the role of robots in service industries [2], and skill-learning frameworks for manipulation tasks [3]. Advances in technology, such as AR-assisted deep reinforcement learning [4], proactive collaboration models [5], and adaptive neural network control in physical HRI [6], are enhancing mutual-cognitive capabilities and safety. Moreover, the intersection of cognitive neuroscience and robotics [7] is paving the way for future research directions. Active learning for user profiling [8], garbage image classification [9], and the importance of nonverbal cues [10] further contribute to the personalization and effectiveness of HRI. These studies collectively underscore the importance of human-centered design in the evolution of smart, adaptive, and interactive robotic systems, particularly in the context of Industry 5.0 [11], while also highlighting areas for future exploration, such as user experience evaluation [12] and gesture-based communication [13]. Robots are expected to be socially intelligent, that is, capable of understanding and reacting accordingly to human social and affective clues [14]. A robot with social perception is able to understand the intentions of the users with the purpose of predicting the best way to help them [15]. The scenario of HRI is also extended into muti-agent interaction [16]. It is implemented using robots with cognitive architectures [17,18,19]. The cognition of robots can be established by understanding world models [20] and behavior models [21].

Several issues were raised in previous HRI designs. First, the robots were designed for a specific purpose, which makes the architecture not generic in complex social environments. Furthermore, the scenario of robot application focuses on the process during HRI, while how to detect the start point of communication is seldom discussed. To fill the gap above, the paper aims to develop a comprehensive robotic system, termed the robot cognitive system, through three models: the engagement model, the intention model, and the HRI model, for complex environments. To demonstrate the advantages of our system, our team has developed a novel dual-arm-hand mobile robot. The overall contributions can be summarized as follows:Four layers of HRI are proposed that organize behaviors between humans and robots and elucidate the significance of action and engagement.A robot cognitive system that empowers the robot to identify interactors in complex environments and provide suitable responses is constructed using existing learning-based models and the integration of an improved Hidden Markov Model (HMM).This paper also proposes the engagement comfort index and the naturalness index. The engagement comfort index is segmented into four stages of engagement to deduce the suitable moment to initiate a conversation based on head pose and eye movements. The naturalness index detects a user’s feedback, allowing the robot to take appropriate actions.To enhance the mobility and functionality of robots, we constructed a novel mobile robot equipped with hands and arms, demonstrating our framework’s efficacy and an authentic HRI sensation in complex environments.

## 2. Background

### 2.1. Human–Robot Interaction

Human behaviors, such as speech, facial expressions, actions, and gaze direction [22], are all potentially significant behavioral cues for robots. These cues provide direct and intuitive messages for engaging in interactions. Other dimensions of human behavior, including emotions [23], intentions [24], and social space [25], are implicit, rendering the observer potentially unaware. Additionally, humans perceive how robots affect their feelings, for instance, through the maintenance of socially acceptable distances and gaze behaviors during interactions. Factors including the velocity and direction of approach, as well as speech characteristics (tone, speed, language, etc.), also play crucial roles in influencing human comfort during social interactions.

According to [26], humans derive four levels of benefits from robots: safety, comfort, naturalness, and sociability (see Figure 1). Adhering to these principles, HRI can be categorized into two approaches: user-centric HRI and robot-centric HRI. User-centric HRI is designed from the human perspective, emphasizing the robot’s provision of superior service to humans. Conversely, robot-centric HRI focuses on offering humans appropriate policies from the robot’s perspective.

Generally, research on HRI can be categorized into two approaches based on the research design methodology employed [27]:**User-centric HRI.** The objective of the interaction is to ensure a high-quality user experience by enabling robots to fulfill the desired objectives of humans.**Robot-centric HRI.** The interaction aims to design robots that can perceive, cognize and act effectively with their surroundings.

The design of HRI varies, with different approaches favored depending on the objective. For instance, when the goal is to listen to a person, HRI design must prioritize the observation of human behaviors, employing a robot-centric approach. Conversely, a service robot, such as a sweeping robot, is expected to optimize its path to efficiently accomplish tasks assigned by humans. Therefore, a proficient design must simultaneously accommodate a multitude of objectives and challenges. Moreover, it is imperative that robots are endowed with the capability to understand the user’s emotions, intentions, and beliefs, including those communicated through bodily and vocal cues, in addition to those explicitly expressed.

### 2.2. Engagement

Several approaches were developed for directly measuring engagement, including questionnaires, interviews, discussions, and heart rate monitoring. These methods record human behavior and interaction through the involvement of certain tools, such as heart rate monitors. Brockmyer et al. [28] developed the Game Engagement Questionnaire to enhance users’ engagement. Monkaresi et al. [29] combined heart rate monitoring with facial tracker features to improve engagement detection. However, to observe engagement without human intermediaries, methods oriented toward objectivity are required [30]. By directly observing human features, such as speech, body movement and gestures, and facial expressions, robots can become more human-like, and human needs can be detected at any time. For example, Youssef et al. [31] utilized features, including gaze, head, face, and speech, to detect user engagement breakdown during human–humanoid interaction.

For social robots, the primary challenge lies in identifying the moment of engagement and maintaining it without interruption. Indeed, robots must be capable of recognizing the state of engagement during HRI and discerning the cessation of engagement (disengagement) when the interaction is actively terminated. Drawing on previous research, we propose an HRI model that incorporates an engagement comfort index, which relies on observing behavioral cues to detect the four states of engagement. Our model includes action recognition, eye gaze, and head pose as integral components, providing intuitive information essential for the engagement system. 

**Eye gaze.** Human beings do not maintain constant eye contact during face-to-face conversations, particularly at the beginning or end of an engagement. Each individual exhibits distinct behaviors while speaking, such as looking downward in contemplation or incorporating gestures.**Action.** Actions involve a human either handing over or receiving an item, which can be seen as maintaining engagement through physical activity. Throughout these processes, the focus is primarily on the object rather than the individual. Consequently, it is imperative to consider actions concurrently during the interaction.**Head pose.** Head poses serve as an approximation of eye gaze. When the target face is obscured, direct reception of messages from the eyes is not possible; instead, information is inferred from the head pose.

## 3. Robot Cognitive Architecture

When engaging in interactions with others, it becomes imperative to infer their intentions by interpreting peripheral signals. HMM is a time-serial model that makes decisions based on hidden states in the past. The engaging signals of humans vary between individuals, which causes problems when considering an explicit model to calculate human intention. To guess the intention through the unobvious states, HMM can be implemented wherein the states represent an individual’s intentions, and the observations encompass informative signals such as actions, eye gaze, emotions, and distance.

In practical applications, the robot cognitive system was constructed with two stages: the engagement stage and the intention stage. Subsequently, the HRI model is employed to deliver specific feedback. Given that our robot operates in complex environments, the engagement model detects whether an individual in the environment intends to interact with it at an early stage. Upon identifying the interactor, the intention model is activated to ascertain their intentions, thereby enabling the robot to provide more effective feedback. 

The subsequent content will introduce the main algorithms in the first part, followed by detailed presentations of the two architectures in the second and third parts, respectively. Ultimately, the HRI model is presented. Figure 2 illustrates the robot’s cognitive system as discussed in this paper.

### 3.1. Engagement Model

Before a conversation starts between two individuals, several gestural signals are observed such as looking at, facing, and turning their body toward others. Each of these signals indicated the possibility of engagement. In our engagement model, we focus on two states: engagement and disengagement, which are determined through three observations: eye gaze, head pose, and actions. Learning-based methods are employed in our architecture for eye gaze and head pose to facilitate an end-to-end system. For action recognition, we utilize the SlowFast model [32], capable of achieving real-time recognition on a mobile robot. In this section, a comprehensive explanation of the overall architecture and the model will be presented. Before delving into details, a brief overview of each observation method is provided as follows.

**People detector and face detector for head pose estimation and eye tracking.** YOLOv3 is well known for recognizing objects from a complex background with robust performance. The model calculates the images and provides labels with bounding boxes (frame u, frame v, height, width). Before implementing the engagement model, raw images captured by the camera are initially fed into the YOLOv3 multiple people detection system to track each individual by dynamically allocating an identification (ID). Upon determining the location and ID of a person in the image, the face detection algorithm utilizing OpenCV-dlib is employed to extract the facial area. This is significant because YOLOv3 is capable of recognizing individuals even when they are not facing the camera. The face extractor must be integrated to determine whether it captures facial features in the given image. Subsequently, the extracted images are input into two models—head pose estimation and eye tracking—to obtain the angles. For detecting multiple individuals, FairMOT [33] is implemented within our architecture.**Action recognition**. The identification of engagement must be complemented by action, as certain behaviors, such as handshaking, are considered indicative of engagement. Therefore, to facilitate real-time action detection, the SlowFast model was incorporated into our engagement model.**Engagement HMM.** In the engagement HMM, the aforementioned results from each observation are input into the engagement model, which integrates both the sliding window and the moving average techniques. Based on the observations, this model subsequently decodes and predicts the most probable state.

#### 3.1.1. FSA-Net in Head Pose Estimation

In our architecture, the detection of individuals using YOLOv3 is initially implemented. Subsequently, a learning-based face detection algorithm employing the Maximum-Margin Object Detector (MMOD) [34] is applied for face recognition on OpenCV-dlib with GPU support. The equation for MMOD is expressed as follows:(1)$$\underset{w,\xi }{\min}\frac{{\left\|w\right\|}^{2}}{2}+\frac{C}{n}\sum_{i=1}^{n}\xi_{i}\;s.t.\;F\left(x_{i},y_{i}\right)\geq \underset{y\in Y}{\mathrm{ma}}x\;F(x_{i},y)+\Delta (y,y_{i})-\xi_{i},\forall i\;\xi_{i}\geq 0,\;\;\;\forall i$$ where w is the parameter vector, C is the constant parameter usually used in SVM, F is the score function with an input set of images $x_{1},x_{2},x_{3},\ldots ,x_{n}$ and corresponding labels $y_{1},y_{2},y_{3},\ldots ,y_{n}$ and $\xi_{i}$ is the value of the upper bound from the training set $\left(x_{i},y_{i}\right)$. To find the optimal solution of this equation, quadratic programming is used, following [34], to solve this problem.

Finally, head pose estimation is employed subsequent to the derivation of the face region. In the realm of deep learning for head pose estimation, prevalent methods involve extracting key points from target faces and utilizing face models to ascertain the yaw, pitch, and roll angles. For an end-to-end application, FSA-Net is utilized in our engagement model, circumventing the need for landmark or key point detection. Owing to the original face detector—OpenCV cascade classifier in FSA-Net—exhibiting low accuracy, MMOD with OpenCV-dlib is implemented to enhance the accuracy of the results. The experimental sections present a benchmarking comparison of these methodologies.

To classify whether the agent is facing the camera or not, we set up the constraint function for each angle—yaw (α), pitch (β), and roll (γ):(2)$$f(\alpha ,\;\beta ,\gamma )=\left\{\begin{array}{cc} 1, & \left\|\alpha \right\|\leq \alpha^{\prime }\cap \left\|\beta \right\|\leq \beta^{\prime }\cap \left\|\gamma \right\|\leq \gamma^{\prime }\\ 0, & else \end{array}\right.$$

#### 3.1.2. CNN-Based Eye Tracker

The system can be divided into two components, as illustrated in Figure 3. The initial segment utilizes the input image obtained from the OpenCV-dlib face detector to identify the eye regions through an eye gaze detector. These data are subsequently conveyed to the second segment of the system, a deep learning model that determines the direction of each individual’s eyes. Within the CNN-based classifier, a previous pure CNN model [35] is employed to ascertain the output angle. However, when the input image of the eye region is blurred or small, SRWGAN, developed in [35], is integrated into our framework.

Each person’s eyes can be represented by a 3D model to acquire angles. To check whether the agent looks at the robot, we similarly use the constraint function as follows:(3)$$g\left(\theta_{\alpha },\theta_{\beta }\right)=\left\{\begin{array}{cc} 1, & \left\|\theta_{\alpha }\right\|\leq {\theta^{\prime }}_{\alpha }\cap \left\|\theta_{\beta }\right\|\leq {\theta^{\prime }}_{\beta }\\ 0, & else \end{array}\right.$$
where $\theta_{\alpha }$ is the horizontal angle and $\theta_{\beta }$ is the vertical angle. The $\theta_{\alpha }^{\prime }$ and $\theta_{\beta }^{\prime }$ is the constraint value for this equation.

#### 3.1.3. Action Recognition in Engagement

For the real-time demonstration, the learning-based architecture known as SlowFast is utilized for action recognition. To extract information pertaining to both human–object interactions, such as washing dishes, and human–human interactions, such as shaking hands, the Kinetics-400 dataset is employed to generate our pre-trained model. From the 400 categories, we categorize them into interactions involving humans and those not involving humans. The interactive behavior is presented in Table 1. The subsequent formulation represents our action model.
(4)$$h(a)=\left\{\begin{array}{cc} 1, & \mathrm{interaction}\;\mathrm{behaviors}\\ 0, & \mathrm{else} \end{array}\right.$$

#### 3.1.4. Engagement HMM

In engagement HMM, we formulate previous parameters to generate the prediction of engagement. For the multiple people $P=\left\{p_{1},p_{2},p_{3},\ldots ,p_{n}\right\}$, each person owns their HMM model $M=\left\{m_{1},m_{2},m_{3},\ldots ,m_{n}\right\}$. For the input value from the head pose, eye angles, and action, a vector of three properties is encoded by a label encoder Γ(·) to derive the output σ:(5)$$\sigma =\Gamma (\left[\begin{array}{ccc} f(\alpha ,\beta ,\gamma ) & g(\theta_{\alpha },\theta_{\beta }) & h(a) \end{array}\right])$$

For the estimation of head pose and eye gaze, our primary concern is whether agents are facing the camera and looking into the lens, denoted by F and L, respectively. Furthermore, actions within our HMM are denoted by A, and it is essential to determine whether these actions represent interaction behaviors for each agent. The emission and transition probability matrices are presented as follows:(6)$$B_{e}={\left(b_{ij}\right)}_{n\times m}$$
(7)$$A_{e}={\left(a_{ij}\right)}_{n\times m}$$
where n is the number of states and m is the number of observations. In the paper, we use two states—engagement and disengagement—and eight observations from the above three conditions ($2^{3}$). The initial value of the matrix is chosen empirically and updated by the Baum–Welch algorithm, which can find the maximum likelihood estimate of the parameters. The initial probability π, in which we consider there is a high probability of starting at the disengagement state, is given as:(8)$$\pi \in N_{1\times n}$$

Algorithm 1 shows the engagement model for multiple people. For the input with raw image, initial probability π, transition probability $A_{e},$ and emission probability $B_{e}$, the output result is a people list with a new state. For step 4, agent IDs and bounding boxes are extracted by YOLOv3, and this continues until IDs are empty sets. Next, we check whether the detected agent is in the P list or not. If $id\in id_{use}$ (in step 6), the existing person object is obtained. Otherwise, it creates a new person object for the P list. From step 14 to step 16, we achieve the observation using the methods described earlier. The label encoder is used to generate the observation in step 17. Finally, the Viterbi algorithm is used to acquire the most likely path of states.

**Algorithm 1**. Engagement HMM for Multiple People**Input:**img,π,Ae,Be**Result:** new people list with new state**struct** p**:**id←0
 obs←{}
 state←{}
**end struct**
P←{} empty person list with struct p;$id_{use}\leftarrow \{\}$**repeat**ids,bboxes←YOLOv3(img)**for each** bbox in bboxes **and** id in ids **do****if** $id\in id_{use}$agent←P[id]**else**agent←new struct pagent[id]←idP←P∪agent$id_{use}\leftarrow \;id_{use}\cup id$**end if**α,β,γ←FSA(bbox)$\theta_{\alpha },\theta_{\beta }\leftarrow eye\_tracker(bbox)$a←slowfast(bbox)$obs\leftarrow \Gamma (\left[\begin{array}{ccc} f(\alpha ,\beta ,\gamma ) & g(\theta_{\alpha },\theta_{\beta }) & h(a) \end{array}\right])$agent[obs]←obs∪agent[obs]$state\leftarrow Viterbi\left(\pi ,A_{e},B_{e},agent\left[obs\right]\right)$agent[state]←state**end for**
**until**
ids=∅


### 3.2. Intention Model

John J. Mearsheimer posits that “intentions ultimately reside in the minds of decision-makers, rendering them unobservable and immeasurable”. Such intentions, being ensconced within the minds of decision-makers, elude direct observation and quantification. Nonetheless, emotions can facilitate an observer’s inference of targeted cognition, thereby leveraging its potential intentions. 

Based on the architectural framework, we minimized the number of intentions required for detection, as the concepts of openness and unawareness are encapsulated within the constructs of engagement and disengagement. The sole method to ascertain absence is by recording the elapsed time of non-interaction, which, however, lacks robustness for interactive purposes. At the level of intention, we focus on the intensity of the target’s desire to engage with us, which is critical for starting or stopping an HRI process in a more user-friendly way. The intention is often observed in one’s behavior, such as eye contact, gestures, and emotion. To predict the hidden state between intention and behaviors, HHM is utilized to construct an intention model. Consequently, three sentiment indices—positive, neutral, and negative—are employed for observation within the HMM. These indices reflect the attitude or opinion that an individual expresses in a given post (refer to Table 2).

To map sentiment indices for face-to-face social interactions, we propose seven states of intention, as shown in Table 3. This table includes engagement, emotion, and sentiment to identify the appropriate intention states.

#### 3.2.1. Emotion Classifier

To find out the relationship between emotions and human behaviors, we adopt the facial-feedback hypothesis in [24] because facial expressions are easy to extract from raw images and infer the target’s emotions from this behavioral cue. In the emotion classification, the network constructed is also implemented into our emotion classifier (see Figure 4). FER-2013 [32] is used as our training dataset in this paper. For emotion classifier, ω is denoted from the output of the emotion classifier by following a definition. Then, we regulate it using Table 4 to produce a sentiment value from the emotion, which serves as an input of intention HMM in Section 3.2.3.
(9)$$\omega =\left\{\omega |\omega \in Z,0\leq \omega \leq 7\right\}$$

#### 3.2.2. Speech Analyzer and Google NLP Sentiment

At first, the robot listens to the voice of the agents and pushes it to Speech Analyzer. In Speech Analyzer, the voice is analyzed by Google speech-to-text API which returns the most likely words. Then, we push the words to Google NPL Sentiment, a sequence-to-sequence model based on the pre-training of Deep Bidirectional Transformers (BERT) that can understand the meaning of the text, and the return from Google NLP Sentiment are scored and given a magnitude. The score represents the overall emotions from the text and ranges between −1.0 (negative emotion) and 1.0 (positive emotion). The magnitude m denotes the strength of the emotion from each emotional word (e.g., fear, happy, …) and ranges between 0.0 and infinity. In order to integrate these indicators clearly, we divided them into three parts: neutral (−0.3≤s≤0.3), positive (0.3≤s≤1), and negative *(*−1≤s≤−0.3). To formulate the score and magnitude, we used the following equation:(10)$$s,m=NLP\left(t\right)\;e=\left\{\begin{array}{c} 1,\left\{s\in Q:0.3\leq s\leq 1\right\}\cap m\geq \mathrm{threshold}\\ -1,\;\left\{s\in Q:-1\leq s\leq -0.3\right\}\cap m\geq \mathrm{threshold}\;\;\\ 0,\;\mathrm{else} \end{array}\right.$$

#### 3.2.3. Intention HMM

For speech sentiment, three sentiments—positive, neutral, negative—are integrated into intention HMM. Once the agent does not speak when detected, we consider that the states should be neutral. For sentiment input—emotion, speech, and the robot’s facial expression—we used the following formulation to generate the most likely sentiment using label encoder Γ(∙) in Equation (11).
(11)$$\sigma_{i}=\Gamma (\left[\begin{array}{cc} \omega & e \end{array}\right])$$

To apply for the detection of multiple people, Algorithm 1 is also used in our intention HMM. The main difference is the function of all learning-based models. The emission and transition matrices are shown as follows:(12)$$B_{I}={\left(b_{ij}\right)}_{n\times m}$$
(13)$$A_{I}={\left(a_{ij}\right)}_{n\times m}$$

Then, the initial state starts from neutral, so we derive the initial probability:(14)$$\pi_{I}\in N_{1\times n}$$

### 3.3. Human–Robot Interaction Model

In this section, we explore the integration of HRI during interactive processes. Previous studies have utilized prerecorded transcripts that are activated upon the occurrence of specific events. To enhance robustness, Google Dialogflow was incorporated into our model. This addition facilitates automatic responses to inquiries and streamlines the creation of customized dialogue flows, thereby enhancing the system’s interactive capabilities. Its value lies in addressing the significant challenge of designing scripts for every conceivable scenario in complex environments, particularly when the robot is required to interpret social distance, emotional engagement, and interaction dynamics. Consequently, we developed our dialogue flow using the Google Assistant SDK. Figure 5 illustrates our model’s workflow. Initially, human behaviors, including actions, emotions, and speech (processed via Google’s speech-to-text service), are fed into the HRI Model. This model contains predefined scenarios and responses, with Google Dialogflow generating suitable output text for these conditions, subsequently vocalized through Google’s text-to-speech service. 

To ensure the robot makes the correct decision at the appropriate time, two indices—the engagement comfort index and the naturalness index—are utilized to identify the stage of engagement and user feedback, respectively.

#### 3.3.1. Engagement Comfort Index

The Engagement Comfort Index is utilized to delineate the four stages of engagement: the point of engagement, the period of engagement, disengagement, and reengagement. Prolonged conversation and interaction result in discomfort in HRI, such that engagement is a function of time, which can be represented by the following sigmoid function:(15)$$E\left(t\right)=-\frac{1}{1+e^{-c_{1}(t-c_{2})}}$$

According to eye contact anxiety, interactors should hold eye contact for about four to five seconds at a time. Therefore, we define the parameter with $c_{1}=0.6$ and $c_{2}=5$ as shown in Figure 6. We define the engagement comfort index, which ranges from 0 to −1. A value of engagement comfort index of −1 indicates the interaction has reached an uncomfortable state and the topic must be closed in case of interaction fatigue. If the value of the engagement comfort index reaches −0.231 (about 3.0 s), we assume that the target is willing to interact further with the robot (point of engagement). When the value of the engagement comfort index reaches −0.83 (about 7.5 s), it represents that the target may feel anxious and have eye contact anxiety. The interval between the two values indicates that period of engagement.

#### 3.3.2. Naturalness Index

By comprehending human emotions and sentiments, the concept of naturalness strives to foster a friendly approach in HRI. Consequently, we introduce the naturalness index as a measure of user feedback, positing that a user-centric HRI approach can facilitate the detection of such feedback. $s_{e}$ is calculated using Table 4, and $s_{s}$ is calculated using Equation (10). The naturalness index is derived from the emotion and sentiment scores, and is represented as follows:(16)$$\begin{array}{l} N_{index}=(s_{s}\cdot (1+m_{s}))w_{s}+s_{e}w_{e}\\ N_{index}^{*}=\frac{N_{index}}{\left\|N_{index}\right\|} \end{array}$$
where $s_{s}$ is the sentiment score and $s_{e}$ is the emotion score. $w_{s}$ and $w_{e}$ are the weight of this equation. $m_{s}$ is the magnitude of the sentiments. The naturalness index $N_{index}^{*}$, which represents the level of naturalness during an interaction, ranges from −1 to 1. The negative values represent the interaction as unfriendly and reckless, while positive values are the opposite. The index is used in our paper to evaluate the naturalness of the HRI.

## 4. Implementation

The robot cognitive system can detect an intention to engage and provide a service at appropriate times and is feasible in a complex environment with multiple users. In order to demonstrate our architecture and an authentic HRI sensation, we developed a new mobile robot, Mobi, with dual arms and hands (Figure 7). The robot can express emotion and body gestures vividly with six DOF arms and two DOF hands. Mobi is equipped with an Intel RealSense depth camera D435i, which can collect RGB and depth stream simultaneously. In order to let Mobi adapt to a complex environment, it is equipped with a laser sensor with a 180-degree visibility range, which is used for location and path planning purposes. To control its facial expression, the face is installed with a seven-inch LED touchpad screen to show facial expressions and interact with clients.

### 4.1. Scenario

Figure 8 illustrates the scenario discussed in this paper, wherein the entire experiment is configured in a retail environment. Blue and gray circles symbolize robots and humans, respectively. In this setup, the robot assumes the role of a shop attendant, poised to offer assistance promptly when required by customers. For instance, the individual at the center of the figure expresses a desire to learn about the history of the exhibits. The robot identifies this interest as an anomaly, subsequently initiating an inquiry and providing an explanation. This scenario encompasses three distinct technologies:**Engagement and intention detection.** To identify people in need of help in a complex environment, engagement and intention detection must be used in the system.**Navigation and SLAM.** The robot must know the overall environment and the location of obstacles in order to guide or lead guests.**HRI design.** To make sure the interaction is safe and smooth, HRI must be considered. For example, the walking speed of the robot must change dynamically depending on the environment or obstacles.

### 4.2. Engagement Model

Due to the absence of existing datasets on engagement detection, video footage of interactions was collected using a camera. In the benchmarking test, ten participants were recruited for the experiment. These participants interacted with Mobi, following its instructions, and the experiment was designed to take place within an exhibition setting. Initially, Mobi introduced itself and began to answer questions using Dialogflow (refer to Figure 9). Subsequently, Mobi provided introductions to the exhibits within the area. The experiment aimed to induce a state of engagement in the interactors at specific times; therefore, intervals were labeled, and the average accuracy was calculated. The results of tracking a target through the engagement model are presented in Figure 10, with sampling at 15 frames per second. The white area, indicating no input data, signifies that the model did not identify the target, whereas the green area indicates that the target was engaged.

To validate the reasonability of our engagement model, the F1 score (Equation (17)) is used to verify it. If the bounding box is not identified, we treat it as a false negative (FN).
(17)$$F_{1}\mathrm{score}=2\;\frac{\mathrm{Precision}\cdot \mathrm{Recall}}{\mathrm{Precision}+\mathrm{Recall}}$$

Table 5 presents the outcomes of our experiment. We recruited ten subjects for the experiment. The subjects interact with Mobi followed by its instructions, and we design the experiment in an exhibition. The experiment tests the Robot Cognitive Architecture and whether it can recognize the engagement timing or not. The mean F1 score is 75.7 percent, accompanied by an average interaction time of 21.7 min. Due to the potential failure of our model to identify the subject, Subject 5 is marked as unrecognized, resulting in an F1 score of zero. Throughout the interaction, individuals exhibited varied forms of engagement. Some uttered a word and departed, whereas others demonstrated significant curiosity.

### 4.3. Engagement Model Benchmark

For emission and transition probability, the value of each entry is shown below.
(18)$$B_{e}={\left(b_{ij}\right)}_{n\times m}=$$



L’F’A’L’F’AL’FA’L’FALF’A’LF’ALFA’LFA
Disengagement

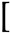

0.60.10.100.10.050.050

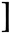

Engagement0.00.050.10.10.050.10.30.3


(19)
$$A_{e}={\left(a_{ij}\right)}_{n\times m}=$$






Disengagement

Engagement
Disengagement

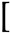



0.8



0.2



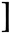

Engagement

0.2



0.8



The L,F,and A symbols represent whether the target is looking at the camera, is facing the camera, or is demonstrating interactive behavior, while $L^{\prime },F^{\prime },and\;A^{\prime }$ represent the opposite. For the initial state, we consider there is a high probability that the initial state will be disengaged. The initial probability is written below.
(20)$$\pi ={\left[\begin{array}{cc} 0.8 & 0.2 \end{array}\right]}^{T}$$

The thresholds are set below to check whether the target is facing the robot. The threshold of the yaw (α) angle, which is out of the screen, is relatively high because it does not affect the head orientation. The threshold of the eye tracker is set below because of the instability of the eye (blinking or eye movement).
(21)$$E(t)=-\frac{1}{1+e^{-c_{1}(t-c_{2})}}$$

In this experiment, an interactor is able to pose any question regarding the store, encompassing product location, and exhibit introductions. Mobi responds to these inquiries based on the HRI model and Google DialogFlow, upon the engagement comfort index attaining a value of −0.2, which signifies the point of engagement. Figure 11 illustrates the fluctuations in the engagement comfort index throughout the interaction. It was observed that the interactor’s state transitions from engagement to disengagement when Mobi initiates product introduction, such that a suitable engaging moment is found. Concurrently, as the target redirected their gaze towards Mobi, a decline in the engagement comfort index was noted.

### 4.4. System Integration

In this demonstration, Mobi functions as a shop assistant, providing intelligent voice services and guidance. Mobi is capable of engaging in everyday conversation, including information about the current time and place, among other topics. Customers in this scenario can inquire about exhibitions and product locations. We developed a system that integrates information from input images, speech, and maps simultaneously. Initially, speech and the engagement comfort index are continuously monitored. Speech is processed using a Speech-to-Text model to generate request text, which is then input into DialogFlow to produce the response text. Subsequently, the HRI model utilizes the engagement comfort index and response text to determine the location of the sought object and activates Text-To-Speech to generate the corresponding sound wave. Based on the location, our SLAM and navigation system directs Mobi to the appropriate location. Initially, Mobi identifies individuals willing to interact with it through the engagement comfort index (refer to Figure 12 and Figure 13). If the index exceeds a certain threshold, Mobi initiates interaction by greeting the customer and offering assistance. Mobi responds to questions and takes actions based on Dialogflow outputs.

Figure 14 illustrates the navigation scenario. Should the interactor wish to locate a product, Mobi guides the customer to the specific item utilizing our system. Following Mobi’s instructions, the subject can successfully locate the product within the store. Through our robot’s cognitive system, engagement comfort index, and naturalness index, Mobi is capable of identifying individuals who intend to interact with it in complex environments and of detecting the user’s reactions, including emotions and sentiments.

Compared to other cognitive systems such as SGOMS [36] and Soar [37], our design focuses on the agreeableness and appropriate timing of HRI rather than a problem-solving agent. It shares a similar architecture with the feature-based behavior model [38] to distinguish whether the user will engage or disengage, but the impact of emotion recognition from facial expression and speech is adjusted. Instead of modeling human cognitive behavior [39,40,41], our design is an implementable and generic cognitive system for robots, which is able to detect emotions and intentions and provide service with appropriate timing.

## 5. Conclusions

This paper aims to enhance the naturalness and comfort of HRIs. A cognitive HRI framework comprising four layers is proposed to facilitate interactions between humans and robots. Through the design of a robot-centric HRI, several models are employed to generate engagement and intention models, enabling the self-fabricated Mobi robot to detect human behaviors in complex environments. The HRI model allows Mobi to respond appropriately based on the user’s emotions, intentions, and speech, a capability that is implemented and verified in a purchasing scenario. While intentions are crucial, there is no definitive answer during interactions in this research. Incorporating more learning-based models, such as reinforcement learning, into the policy decision model could enhance the robots’ decision-making accuracy, enabling them to navigate complex environments effectively. It was demonstrated that a robot equipped with a cognitive architecture can integrate into social environments and provide services that offer user-friendly experiences.

## Figures and Tables

**Figure 1 sensors-24-03311-f001:**
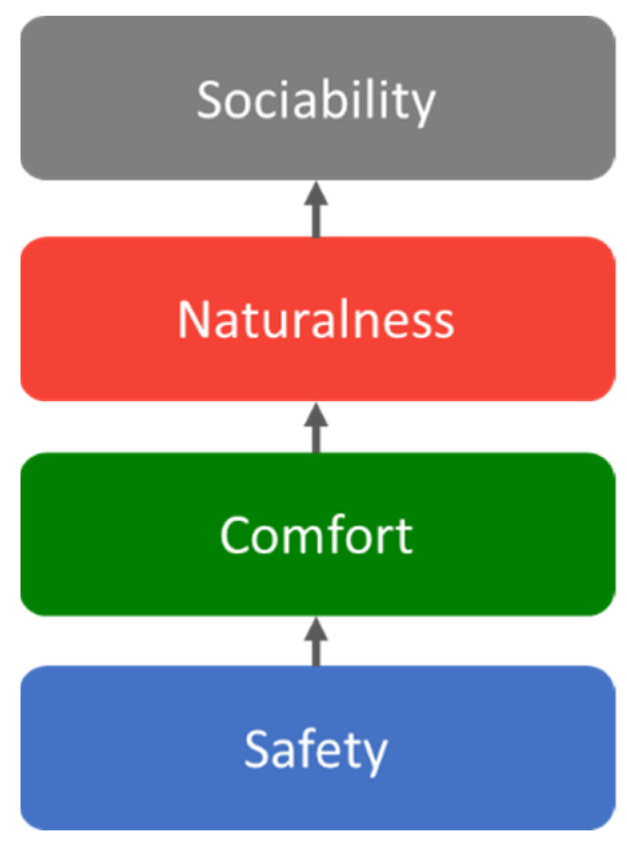
The four layers in HRI. The design of robots should satisfy all design conditions from safety to sociability.

**Figure 2 sensors-24-03311-f002:**
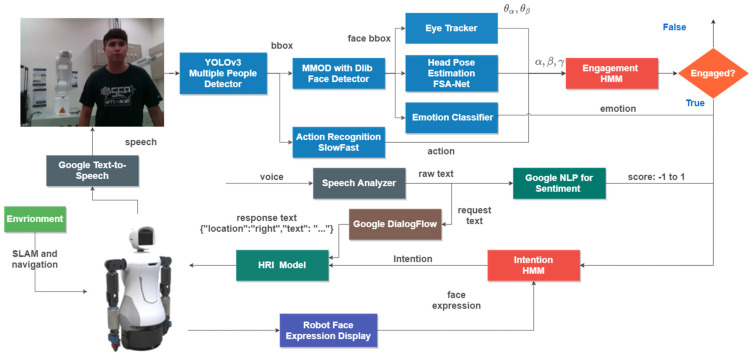
Robot cognitive system. The system can be divided into three parts: engagement model, intention model, and HRI model. In engagement model, head poses, eye angles, and actions are analyzed by engagement HMM to generate appropriate state of engagement. After acquiring the output from engagement HMM, intention model is utilized to identify the intention state of targets. To provide better interaction experience, HRI model is included as a policy system of HRI.

**Figure 3 sensors-24-03311-f003:**
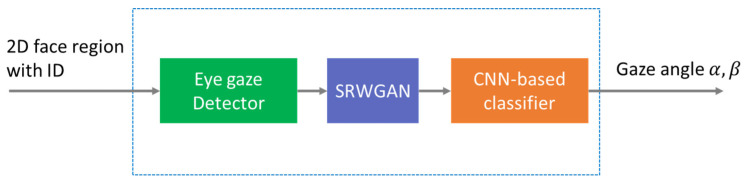
Structure of the gaze tracker. At first, the 2D eye region is processed by eye gaze detector. To restore the eye region image, SRWGAN is included in our model. Finally, a CNN-based classifier is trained to obtain the eye angles.

**Figure 4 sensors-24-03311-f004:**
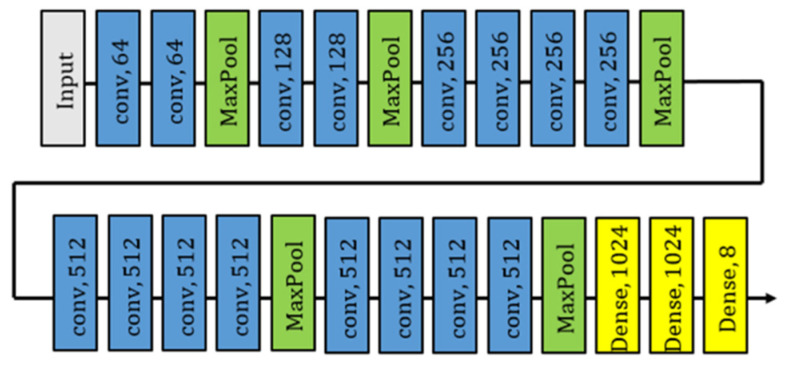
Model of emotion classifier.

**Figure 5 sensors-24-03311-f005:**
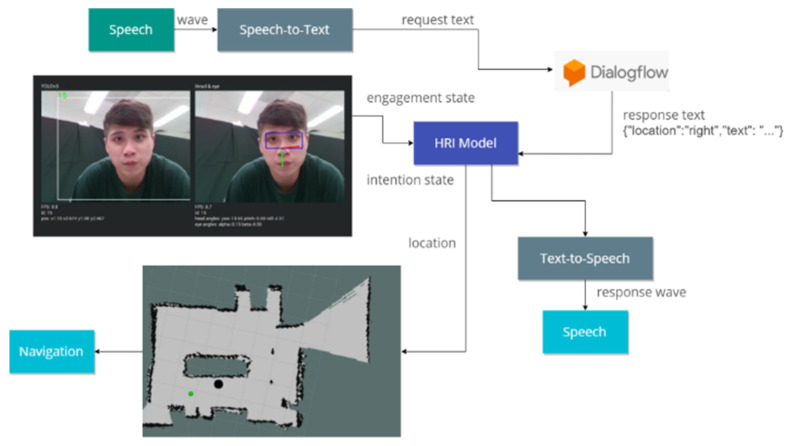
HRI model with google DialogFlow.

**Figure 6 sensors-24-03311-f006:**
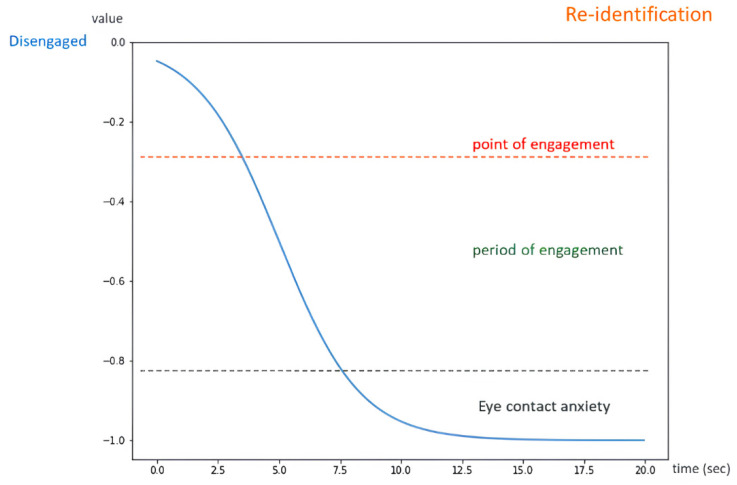
Engagement equation.

**Figure 7 sensors-24-03311-f007:**
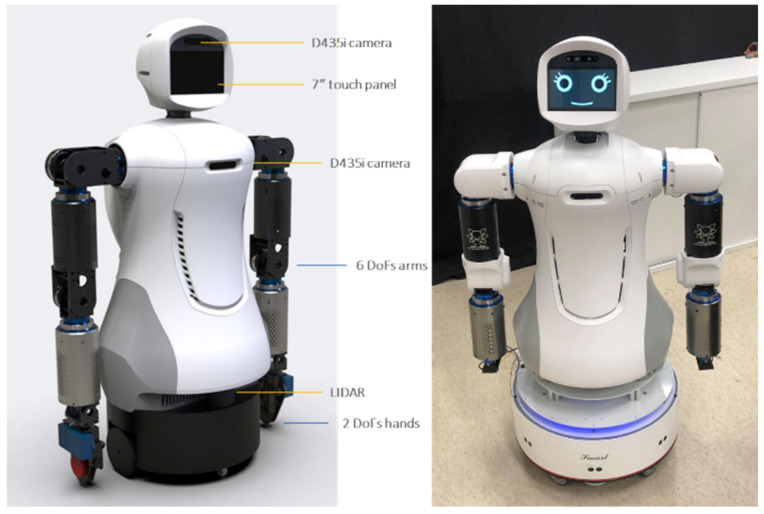
The hardware of Mobi.

**Figure 8 sensors-24-03311-f008:**
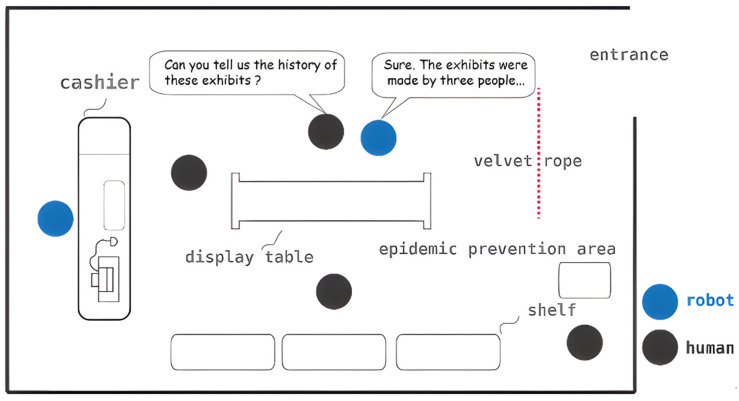
Scenario of our experiment.

**Figure 9 sensors-24-03311-f009:**
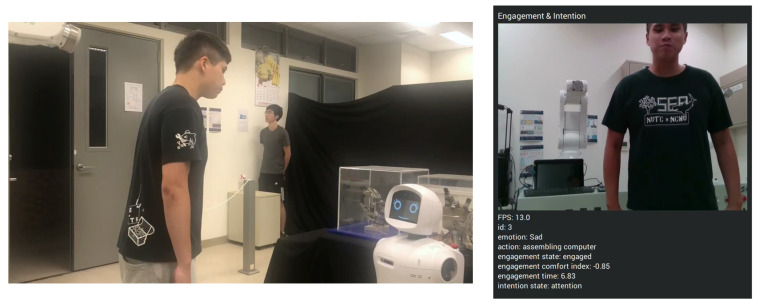
The people interact with Mobi.

**Figure 10 sensors-24-03311-f010:**
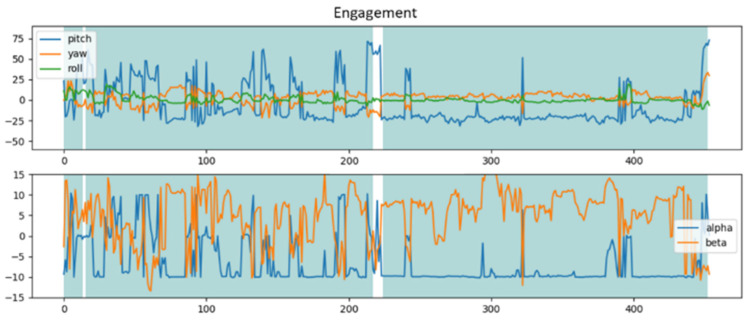
The engagement state of subject.

**Figure 11 sensors-24-03311-f011:**
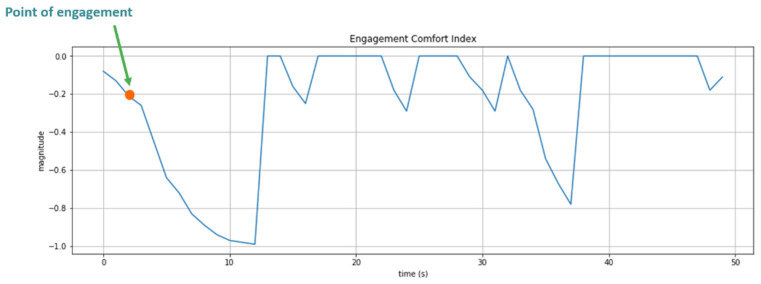
Engagement comfort index changes with time. At 2 s, the index reaches −0.2 such that the start of HRI is detected. Before 10 s, the subject continues staring at Mobi, making the index keep declining to −1.0.

**Figure 12 sensors-24-03311-f012:**
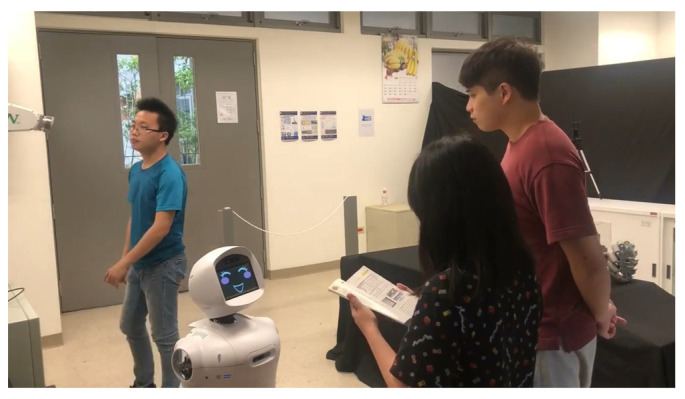
The subjects interact with Mobi.

**Figure 13 sensors-24-03311-f013:**
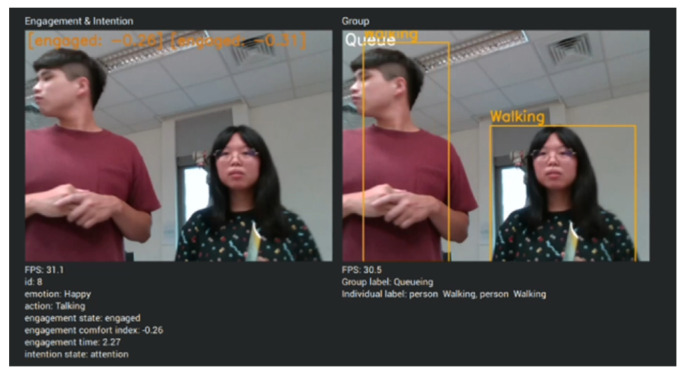
Mobi identified the interactors using engagement comfort index.

**Figure 14 sensors-24-03311-f014:**
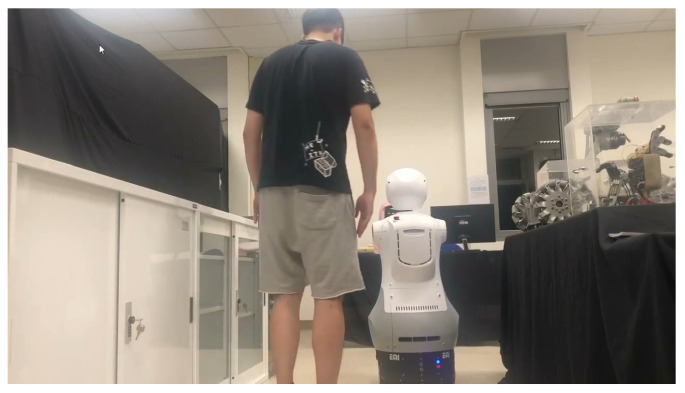
Mobi helped the customer to find the goods.

**Table 1 sensors-24-03311-t001:** Interaction behaviors in Kinectics-400.

ID	Interaction Behavior
2	answering questions
3	applauding
397	yawning
257	pumping gas
279	rock scissors paper
304	singing
319	sneezing
320	sniffing
331	sticking tongue out
344	swinging on something

**Table 2 sensors-24-03311-t002:** Sentiment indexes of HRI.

Index	Meaning
Positive	The person is positive to interact with you and willing to share.
Neutral	The person is neutral.
Negative	The person has no desire to interact with you and potentially needs help.

**Table 3 sensors-24-03311-t003:** States of human beings observed using eye gaze and emotions.

State	Engagement	Emotion	Sentiment
Unawareness	Disengagement		
Openness	Engagement		
Attention	Engagement	Neutral	Positive Neutral
Reflection	Engagement	Disgust Neutral	Neutral
Addressing	Engagement (head pose)	Happy Angry Neutral	Neutral
Empathy	Engagement	Sad Neutral	Negative
Appealing	Engagement	Disgust Sad Angry Fear	Negative

**Table 4 sensors-24-03311-t004:** Sentiment indexes of emotion.

Index	$\mathrm{Emotion}\;(s_{e}$)
Positive	Happy (0.9)
Neutral	Neutral (−0.1), Surprise (0.3)
Negative	Disgust (−0.9), Angry (−0.9), Fear (−0.8), Sad (−0.9)

**Table 5 sensors-24-03311-t005:** Model performances of experiments.

Interactions	Interaction Time	F1 Score (%)
S1	20	98.5
S2	24	92.2
S3	10	84.6
S4	8	92.5
S5	unrecognized	0
S6	36	54.4
S7	17	91.2
S8	34	72.7
S9	18	86.6
S10	28	84.6
Average	21.7	75.7

## Data Availability

Data are contained within the article.

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
