# Peer review of "Implementation of Engagement Detection for Human–Robot Interaction in Complex Environments"

_sensors, 2024, doi:10.3390/s24113311_

Round 1

Reviewer 1 Report

Comments and Suggestions for Authors

The paper presents a compelling exploration of a robotic cognitive system integrating engagement detection to enhance the naturalness of human-robot interactions (HRI). While the topic is interesting and addresses a significant challenge in robotics, there are several concerns within the manuscript that necessitate attention.

Firstly, the introduction lacks a coherent narrative, leaving the gap, challenge, and necessity of the research unclear. Providing a clearer delineation of these aspects would enhance the reader's understanding and appreciation of the study's significance.

Additionally, the manuscript suffers from scattered ideas and methodologies without adequate justification. For instance, the rationale behind the choice of YOLO3 for detection and HMM for modelling engagement remains unaddressed. Clarifying these decisions would strengthen the theoretical underpinning of the study.

Furthermore, the repeated definition of abbreviations such as HMM and HRI in the text indicates a need for consistency and streamlining of terminology usage.

Typos in formulas 6 to 10 and 12 to 14 detract from the professionalism of the manuscript and should be corrected for clarity and accuracy.

The claim of human behaviour and intention prediction seems ambitious, considering the predominant focus on emotion detection. A clearer delineation of these concepts and their operationalization within the study would improve the clarity and rigour of the methodology.

Moreover, the implementation section lacks a thorough analysis of results, leaving the reader without insight into the efficacy and robustness of the proposed approach. Additionally, the assertion in the abstract regarding the system's ability to function in complex environments is not substantiated in the results section, necessitating further validation and analysis.

Comments on the Quality of English Language

-

Author Response

They reply of revision is attached.

Reviewer 2 Report

Comments and Suggestions for Authors

Implementation of Engagement Detection on Human-Robot Interaction in Complex Environments

This manuscript presents a robot cognitive system aimed at improving human-robot interactions in complex environments. The system integrates engagement, intention, and human-robot interaction models, enabling robots to detect human behaviors, intentions, and emotions accurately. The system, demonstrated with the Mobi robot, enhances user experiences, and was validated in a retail scenario.

Authors:

Overall the paper is well written but following suggestion are required to improve it .

1. Ensure the same citation format across the manuscript, either use “,” or “-“while citing two or more paper as reference . It is suggested that the citation style and format of the manuscript should be carefully checked again.

2. Improve the writing of the manuscript. For example, “The design of HRI varies, with different approaches favored depending on the objective," the text contains instances of passive voice, which can make sentences less engaging. Consider rewriting these sentences in active voice for greater clarity and impact. It is suggested that to read the manuscript carefully to improve the overall writing.

3. In section 3 (Robot Cognitive Architecture) i.e "In our engagement model, we focus on two states: engagement and disengagement, which are determined through three observations: eye gaze, head pose, and action." (Lines 163-164) and "Learning-based methods are employed in our architecture for eye gaze and head pose to facilitate an end-to-end system. For action recognition, we utilize the SlowFast model [35], capable of achieving real-time recognition on a mobile robot." (Lines 166-167).
Both of these sentences describe the components and methods used in the engagement model without providing additional insights to support these statements. Emphasize why author prefer to use SlowFast model in this specific scenario. This will help the readers to understand it more clearly.

4. In section 3.1.4, there is missing reference. Read it carefully if there is need, add appropriate reference, for example “From the 400 categories, we categorize them into interactions involving humans and those not involving humans. The interactive behavior is presented in Table I”.

5. Provide more details in Section 3.2.1, on how the emotion classifier works and how it is trained using the FER-2013 dataset. Explain the significance of the sentiment indexes in Table IV and how they are used in the emotion classification process

6. Provide more detailed explanation of how the speech analyzer and Google NLP sentiment analysis are integrated. Explain how the score and magnitude from Google NLP sentiment are used to determine the overall emotional content of the text.

7. Figure 6 and Figure 8 needs to be revised. Its resolution is not up to the journal’s standards.

Comments on the Quality of English Language

 Moderate editing of English language required.

Author Response

They reply of revision is attached.

Reviewer 3 Report

Comments and Suggestions for Authors

1. Clarity and Depth of Theoretical Framework:

- The manuscript needs a clearer articulation of the theoretical foundations underpinning the proposed models. Specifically, how do these models advance our understanding of HRI, and what theoretical gaps are they addressing?

    - A more detailed comparison with existing frameworks and models in HRI literature would help highlight the novelty and significance of the proposed approach.

2. Methodological Details and Validation:

    - The engagement model's criteria for determining the optimal moment for interaction initiation need to be elaborated upon. What specific parameters define "optimal," and how were these parameters validated?

    - The validation scenario presented is limited to a retail environment. For robustness, additional testing scenarios across diverse environments should be included. This would help in demonstrating the system's versatility and effectiveness in varying contexts.

3. Performance Evaluation Metrics:

    - The manuscript briefly mentions the system's performance but lacks detailed evaluation metrics. What specific metrics were used to assess the accuracy, efficiency, and user satisfaction of the interaction? Providing quantitative results and statistical analysis will enhance the paper's rigor.

    - A comparative analysis with baseline or existing systems on similar performance metrics could significantly strengthen the argument for the proposed system's effectiveness.

4. User Experience Assessment:

    - The assessment of the user experience appears to be cursory. Detailed methodologies on how user feedback was collected and analyzed are necessary. Were there any surveys, interviews, or observational studies conducted? How do these findings support the system's claimed improvements in naturalness and comfort of HRI?

5. Technical Descriptions and Clarity:

    - The technical descriptions of the engagement model, intention model, and the overall robotic cognitive system require simplification and clarity. Incorporating diagrams or conceptual models might aid in better understanding the system's architecture and data flow.

    - The manuscript would benefit from proofreading to correct typographical errors and improve the overall fluency and readability of the text.

The manuscript presents an ambitious and potentially impactful approach to enhancing HRI in complex environments. Addressing the above concerns with detailed revisions will significantly improve the paper's clarity, depth, and contribution to the field of HRI.

Author Response

They reply of revision is attached.

Reviewer 4 Report

Comments and Suggestions for Authors

The authors have produced a very interesting research paper on human-robot interaction in complex environments. The way how they developed the framework to enable robots detecting human behaviours, intentions, and emotions via an engagement model, an interaction model, and a HRI models I also noticeable.

I do not have much more comments on the background, framework and literature review presented by the authors. I do have, however, few questions I would be interested authors to answer:

1.       Formatting and layout: please, check equations’ formatting. For instance, Equations (6), (7), and (8) appear to have the belonging set missing. Similarly, subindexes of Eq. (15) look oversized.

2.       Please check Equation (1). It looks to me that what appears to be a subindex of the norm of w is actually its square.

3.       I do not really understand how the Naturalness Index works. Is it a vector? A complex number? If it is neither of these, then it would be impossible for the index to be zero based on the equation (16) (assuming a real space). Furthermore, the range shown in Figure 11 is between 0 and  -1 only.

4.       Following my observation above, Figure 11 needs further explanation.

5.       I’m having concerns on the F1 score and how it was implemented. Could the authors expand about the calculations performed? (as not enough information was provided).

Author Response

They reply of revision is attached.

Round 2

Reviewer 1 Report

Comments and Suggestions for Authors

I don't see much improvement in response to my initial comments: 

1-The manuscript still lacks a coherent narrative.

2-The justification for using YOLO3 and HMM remains unconvincing. 

3-No further comment.

4-No further comment. 

5-The ambiguity surrounding human behaviour and intention prediction in your manuscript hasn't been adequately addressed.

6-No significant improvement. 

7-In the abstract, you stated, "This study develops a comprehensive robotic system, termed the robot cognitive system, for complex environments". Thus, it's contradictory to suggest that this will be addressed in future works!

Comments on the Quality of English Language

-

Author Response

The reply is attached below.

Reviewer 3 Report

Comments and Suggestions for Authors

Author Response

The reply is attached below.

Reviewer 4 Report

Comments and Suggestions for Authors

Thanks to the authors for taking the time to review my comments and modify the research paper accordingly.

No further comments from my side.

Author Response

The reply is attached below.
